# Malignant Hyperthermia in PICU—From Diagnosis to Treatment in the Light of Up-to-Date Knowledge

**DOI:** 10.3390/children9111692

**Published:** 2022-11-04

**Authors:** Martina Klincová, Dagmar Štěpánková, Ivana Schröderová, Eva Klabusayová, Petr Štourač

**Affiliations:** 1Academic Centre for Malignant Hyperthermia, Masaryk University, Kamenice 5, 625 00 Brno, Czech Republic; 2Department of Pediatric Anesthesiology and Intensive Care Medicine, University Hospital Brno and Faculty of Medicine, Masaryk University, Kamenice 5, 625 00 Brno, Czech Republic; 3Department of Simulation Medicine, Faculty of Medicine, Masaryk University, Kamenice 5, 625 00 Brno, Czech Republic; 4Department of Anesthesiology and Intensive Care, St. Anne’s University Hospital and Faculty of Medicine, Masaryk University, Kamenice 5, 625 00 Brno, Czech Republic

**Keywords:** malignant hyperthermia, children, PICU, critical, triggers, sevoflurane, succinylcholine, dantrolene, cooling, *RYR1*

## Abstract

Malignant Hyperthermia (MH) is a rare, hereditary, life-threatening disease triggered by volatile anesthetics and succinylcholine. Rarely, MH can occur after non-pharmacological triggers too. MH was detected more often in children and young adults, which makes this topic very important for every pediatric specialist, both anesthesiologists and intensivists. MH crisis is a life-threatening severe hypermetabolic whole-body reaction. Triggers of MH are used in pediatric intensive care unit (PICU) as well, volatile anesthetics in difficult sedation, status asthmaticus or epilepticus, and succinylcholine still sometimes in airway management. Recrudescence or delayed onset of MH crisis hours after anesthesia was previously described. MH can also be a cause of rhabdomyolysis and hyperpyrexia in the PICU. In addition, patients with neuromuscular diseases are often admitted to PICU and they might be at risk for MH. The most typical symptoms of MH are hypercapnia, tachycardia, hyperthermia, and muscle rigidity. Thinking of the MH as the possible cause of deterioration of a patient’s clinical condition is the key to early diagnosis and treatment. The sooner the correct treatment is commenced, the better patient´s outcome. This narrative review article aims to summarize current knowledge and guidelines about recognition, treatment, and further management of MH in PICU.

## 1. Introduction

Malignant Hyperthermia (MH) is a rare, hereditary, life-threatening pharmacogenetic disease triggered by some commonly used anesthetics—specifically all halogenated volatile anesthetics and the depolarizing muscle relaxant succinylcholine [1]. According to the current state of knowledge, an indisputable role in the pathophysiology of MH is played by the ryanodine receptor RYR1 on the sarcoplasmic reticulum membrane and the voltage-dependent calcium channel Cav1.1 located in the T-tubular membrane of myocytes [2,3]. The genetic disposition to MH is characterized by an autosomal dominant type of transmission in the *RYR1* and *CACNA1S* genes. Recently, a third gene known to be in a causal relationship with MH has been demonstrated. The *STAC3* gene encodes one of the proteins involved in Excitation—Contraction Coupling [4,5]. The genetic background of MH was identified only in approximately 40–60% of MH susceptible patients differing between countries and geographical areas [6,7,8,9,10,11] and it is the aim of nowadays scientific research to reveal the underlying genetic cause of MH.

MH crisis is a severe hypermetabolic whole-body reaction to the pathologically increased level of calcium in muscle cells released by the triggering agents. Increased cell metabolism results in higher oxygen consumption, excessive CO_2_ production, heat production, acidosis, muscle generalized rigidity, etc. All lead to rhabdomyolysis causing hyperkalemia, myoglobinuria, and acute kidney injury (AKI), and if untreated further, it leads to disseminated intravascular coagulation (DIC), cardiovascular failure and death. The incidence of MH crisis is approximately 1:10,000–1:250,000 anesthesia depending on the population and the chosen anesthetic management [1], but the prevalence of genetic abnormality associated with MH is several times higher [12]. The combined prevalence of all MH causal diagnostic variants in total is 1:2750 [13]. Even though MH crisis mortality is currently significantly lower thanks to the availability of dantrolene [14,15], MH is still considered to be a very serious and life-threatening complication. As MH crisis is well preventable by not administering triggers to MH susceptible individuals, that is why the detection and diagnosis of at-risk patients are so crucial [16,17,18].

Several studies found that children aged 15 or younger comprised more than 50% of all MH reactions [1,19,20]. According to a more recent multicentric study, the median age of all studied MH reactions was 12 years [21]. But MH is not entirely the problem of anesthesiologists only. Nowadays, the MH crisis seems to be more often indolent and insidious [22], sometimes it can develop with delay or recrudesce, already outside of the operation room. Volatile anesthetics and succinylcholine, the triggers of MH, are used in pediatric intensive care unit (PICU) as well. Volatile anesthetics for example in the treatment of refractory and life-threatening status asthmaticus and status epilepticus, and in difficult sedation scenarios (AnaConDa^TM^ system), succinylcholine still sometimes in airway management. MH can also be a cause of rhabdomyolysis and hyperpyrexia in the PICU. Moreover, with the ongoing development, progress and availability of modern molecular genetics diagnostics, there is an increased number of genetically diagnosed MH susceptible (MHS) patients or patients at potential risk of MH with an RYR1 variant of unknown significance, even without any personal or family history of adverse anesthetic events suspected to be MH [23]. Typically, these are patients from neuromuscular or metabolic centers being investigated for myopathy, hypotonia, psychomotor development impairment, failure to thrive, etc. In addition, these risk patients are often admitted to PICU, most commonly either for impairment of their primary disease or syndrome, for the postoperative condition, or for any kind of infection. All of this makes the MH topic very important for every pediatric specialist, not only anesthesiologists.

To the best of our knowledge, this is the first narrative review article devoted to MH from the pediatric intensivist´s point of view. We searched from more than 1100 articles for the relevant ones in the PubMed database to bring together this comprehensive review. It aims to summarize current knowledge and guidelines about recognition, treatment, and further management of MH in the PICU environment. Special attention is dedicated to the differential diagnosis and the recognition of patients at risk considering the most recent information, to fill the knowledge gap, especially among non-anesthesiologists. It is not only dantrolene, which saves lives—early warning, diagnosis, and prompt effective therapies are crucial for MH patients to survive, as proven in China, where the availability of dantrolene is very low and yet they were able to reduce the MH mortality [24].

## 2. Topics of Concern Regarding to MH in PICU

As already mentioned, malignant hyperthermia (MH) can lead to severe, life-threatening crises. MH crisis is usually taken only as an anesthesia complication, but in fact, the triggers of MH are quite often used within intensive care as well. The management of the MH crisis in most cases does not terminate in ORs but continues in the ICUs or PICUs, respectively. The late onset of MH crisis outside of ORs may occur as well. Moreover, non-anesthesia-related MH and MH-like complications can occur, and intensivists must readily treat them.

### 2.1. Triggers of MH

By triggers of MH are usually meant the pharmacological triggers only, but MH can be rarely triggered by non-pharmacological triggers as well.

#### 2.1.1. Pharmacological Triggers

As the pharmacological triggers of MH have been identified succinylcholine and all inhalation (volatile) agents, except nitrous oxide and xenon. 

Not every trigger has the same potency for triggering MH. Succinylcholine enhances the clinical response as the combination of succinylcholine with a volatile agent triggers an earlier reaction compared with a volatile agent alone, and survival rates appear to be lower [25]. In the matter of volatile halogenated agents, various studies have shown differences in potency and the time interval between induction of anesthesia and the development of signs of MH with halothane, enflurane, isoflurane and sevoflurane [26,27,28,29]. Halothane triggers MH faster than sevoflurane (median 20 min vs. median 60 min) [29].

#### 2.1.2. Non-Pharmacological Triggers

Rare cases of MH independent of anesthesia exposure have been reported in responses to vigorous exercise, heat stress, and also emotional stress [30,31,32,33,34,35,36,37]. These episodes were referred as “awake” or “stress-related” MH, but terminology might change in the future [38]. 

### 2.2. Typical Symptoms of MH and Differential Diagnosis

The clinical presentation reflects disturbed calcium homeostasis due to genetic defects and MH triggers. The most typical symptoms of MH are hypercapnia, tachycardia, hyperthermia, and muscle rigidity. 

Hypercapnia, due to inappropriately elevated CO_2_ production (raised end-tidal CO_2_ on capnography, tachypnoea if breathing spontaneously), is known to be the earliest sign of the MH crisis. Furthermore, together with increased O_2_ consumption, lead to mixed respiratory and metabolic acidosis. A drop in the oxygen blood saturation (SpO_2_) usually occurs. Cardiovascular signs include inappropriate tachycardia, cardiac arrhythmias, and unstable blood pressure. Hyperthermia might be pathognomonic, but it develops quite late. Profuse sweating and mottling of skin can be present earlier. Masseter spasms (especially if succinylcholine has been used) and generalized muscle rigidity are the early signs of muscle impairment. The later signs of muscle breakdown are hyperkalemia, elevated blood creatine phosphokinase (CK) and myoglobin levels, and dark-colored or “Coca-Cola” colored urine due to myoglobinuria, leading to acute renal failure and acute kidney injury. If the MH crisis stays untreated, patients die due to severe cardiac arrhythmias and cardiac arrest or multiple organ dysfunction syndrome with disseminated intravascular coagulation (DIC), aggravating into multiple organ failure.

None of the clinical features is specific only to the MH crisis; therefore, the broad differential diagnosis must be held in mind. Several other problems or diagnoses can present similarly, most of them are even much more often in operation rooms (ORs) and PICUs environment (Table 1). However, despite the rarity of the MH crisis, the cornerstone of good patient outcomes is to be aware of this diagnosis and commence treatment as soon as possible. 

A 1994 consensus conference led to the formulation of a set of diagnostic criteria—according to MH clinical grading scale (CGS), commonly known as the Larach score. CGS helps predict the likelihood of an acute MH crisis. There are six indicators of an MH crisis: rigidity, muscle breakdown, respiratory acidosis, temperature increase, cardiac involvement and “other indicators” such as metabolic acidosis and rapid reversal of MH after intravenous administration of dantrolene [41]. 

Nowadays, the clinical presentation of MH crisis seems to be more often indolent and/or insidious than truly fulminant. The more insidious character of MH is most likely due to the lower triggering potency of modern volatile anesthetics, the mitigating effects of several intravenous drugs (non-depolarizing neuromuscular blocking agents, alpha 2 adrenergic receptor agonists, beta-adrenergic blockade) or techniques (neuraxial anesthesia) and the routine use of end-tidal CO_2_ monitoring leading to the early withdrawal of triggering drugs [22].

To conclude, the key to diagnosing an MH crisis is to be aware that it can develop in anybody anytime the triggering agents are administered, or as already mentioned very rarely even without them. Monitoring the end-tidal CO_2_ and temperature is the crucial part of identifying the right diagnosis. Unexplained, unexpected increases in end-tidal CO_2_ (EtCO_2_), heart rate and body temperature are highly suspicious, and this should be sufficient to call the event as MH crisis and start the treatment immediately [17,18]. 

### 2.3. When Can Be MH Met in PICU?

#### 2.3.1. MH Crisis Arising in Anesthesia—Continuing the Commenced Treatment 

The most common situation when an intensivist might witness and treat MH in PICU is when admitting a patient suffering from MH crisis after anesthesia. Most MH crises occur in ORs. Once the MH crisis is recognized, exposition to triggers must be eliminated and the causal treatment with dantrolene initiated as soon as possible [17,39]. Even with early recognition and early adequate management of the MH crisis, the patient´s clinical state might be still critical and the transport to PICU is recommended. Not only that, but despite the well-handled situation and improving the patient´s clinical state after dantrolene administration, approximately 20% of patients develop recrudescence after MH reaction within few hours, mean time from initial reaction to recrudescence was 13 h [42]. The more severe the MH reaction, the more likely is to recrudescence [43]. There are many other consequences and complications resulting from the MH hypermetabolic state, and transferring the patient to the PICU is a natural continuation of the management of the developed MH crisis and further treatment. According to guidelines, monitoring the patient in a critical care environment should be for a minimum of 24 h [17,39].

#### 2.3.2. Late Onset of MH Crisis after Anesthesia

Traditionally, the onset of MH is usually known to be within the first hours after exposure to the triggers, depending on the type of inhaled anesthetics, combination with succinylcholine, age, previous exposition to triggers, etc. [26,29]. On contrary, in some already described MH cases, the onset of the MH crisis was much delayed [44,45,46]. Regardless of whether the delayed onset of the MH crisis is even rarer than MH itself, one must take the possibility of MH into account when dealing with complications after surgery and anesthesia. The increase in EtCO_2_ despite effective and sufficient ventilation and ruling out other possible causes of hypercapnia, e.g., endotracheal tube displacement, is usually the leading sign to the right diagnosis of MH. Clinical or biochemical evidence of rhabdomyolysis (increased postoperative levels of CK and myoglobin, myoglobinuria, hyperkalemia) is also highly suspicious for MH. Other symptoms of MH, such as tachycardia, arrhythmia, blood pressure instability, hyperthermia, and combined acidosis might be misinterpreted as signs of different, on PICU much more common, conditions, e.g., septic shock. Although MH does not represent a significant portion of diagnoses related to hyperthermia, when hyperthermia occurs in children exposed to anesthetic agents, MH should be considered in the differential diagnosis [47].

#### 2.3.3. MH Crisis in PICU Not Related to Administration of General Anesthesia

As already mentioned before, the usage of substances triggering MH is not linked only to anesthesia. 

Inhaled volatile anesthetics have a long tradition of use as hypnotic agents in operating rooms and are gaining traction as sedatives in intensive care units [48]. There are a few situations when volatile anesthetics, nowadays mostly sevoflurane, are used in PICU—as therapies for refractory and life-threatening status asthmaticus, status epilepticus, and high and difficult sedation need scenarios with the usage of the AnaConDa^TM^ system [48,49,50,51,52,53,54,55]. A case of MH event after administering inhalational anesthetics to a girl with life/threatening asthma refractory to standard medication was already published [56]. Succinylcholine itself, the short-reacting depolarizing myorelaxant, may trigger MH events without co-administration with volatile anesthetics [57]. In 1993, the Food and Drug Administration (FDA) published a warning regarding serious adverse effects and fatal or near-fatal cases of cardiac arrest after succinylcholine administration [58], and recommended the use of succinylcholine only for “emergency intubation or instances where immediate securing of the airway is necessary, e.g., laryngospasm, difficult airway, full stomach or intramuscular (IM) use when a suitable vein is inaccessible.” Considering the severe side effects and nowadays broadly available and safer option of rocuronium with sugammadex, the role of succinylcholine in pediatric rapid sequence induction (RSI) is now debatable. However, in some countries, especially outside of North America, the use of succinylcholine in RSI is still not uncommon [59,60].

#### 2.3.4. Rhabdomyolysis or Hyperpyrexia in PICU Potentially Caused by MH

Both rhabdomyolysis and hyperpyrexia are quite common conditions in PICUs. The underlying cause of these conditions is multifactorial, and intensivists must be able to make differential diagnosis (Table 1). The most common cause of hyperpyrexia in PICU is an infection [47]. Rhabdomyolysis is at PICU most commonly connected with sepsis, trauma, and cardiac arrest [61]. However, more rare conditions may occur as well—exertional heat syndrome (EHS), exertional rhabdomyolysis (ERM) and drug-induced hyperthermic syndromes.

EHS and ERM occur more often in patients with known or not yet diagnosed myopathy, especially RYR1-related myopathies, without any pharmacological trigger. The link between MH, EHS and ERM is not well explored yet, but it is reasonable to assume that EHS and ERM patients might be at an increased risk for MH, and vice versa [30,62,63,64,65].

Drug-induced hyperthermic syndromes are caused mostly by psychopharmacological drugs like antidepressants, and neuroleptics but also antibiotics, pain killers, anti-Parkinson drugs, and volatile anesthetics [66]. Besides MH they include, neuroleptic malignant syndrome (NMS), serotonin syndrome (SS), parkinsonism-hyperpyrexia syndrome, intrathecal baclofen withdrawal syndrome, adrenergic stimulation caused by psychedelics (amphetamine, MDMA, cocaine, ketamine, phencyclidine, LSD, psilocybin, mescaline), drugs that uncouple oxidative phosphorylation, and anticholinergic syndrome [67,68]. The clinical features of MH, NMS and SS disorders share some similar presentations, such as altered mental state, increased muscle tone, and autonomic instability (hyperthermia, diaphoresis, tachycardia, hypertension), but they seem to have different genetic backgrounds [69,70]. NMS arises after longer exposure to antidopaminergic medications. SS is caused by excess serotonin levels from serotonergic medication, a combination of them increases the risk [71]. The treatment is to withdraw potential triggering drugs and to provide intensive care, specific treatments may vary (more details in Table 2).

#### 2.3.5. Recognition of Patient at Risk for MH in PICU 

Already fully diagnosed MH susceptible (MHS) patient can also be admitted in PICU for various reasons. According to rareness of MH, it might be infrequent, but in case patient or parents or other family members are announcing the MHS status (already known one or patient currently under MH investigation), this kind of patient must never be exposed to the triggers of MH for the obvious life-threatening risks. 

Sometimes the intensivist might be the first who calls suspicion of MH. Previous complications or deaths in anesthesia in personal or family history, suspicion to or known myopathy, recurrent or idiopathic rhabdomyolysis, idiopathic hyperCKemia, exertional heat stroke, detected non-benign variants in genes causally connected with MH or Excitation-Contraction Coupling (nowadays known *RYR1*, *CACNA1S* or *STAC3*, but might be also other genes in the future). 

Many individuals with myopathy due to RYR1 variants may also have susceptibility to MH [13,73]. Some myopathies are in pathogenic connection with MH, most commonly Central Core Disease (CCD), Multiminicore Disease (MmD), Nemaline Myopathy (NM) or Core-Rod Myopathy, Centro Nuclear Myopathy (CNM), Congenital Fiber Type Disproportion (CFTD), Fetal akinesia, Axial myopathy–late onset, King Denborough syndrome (KDS) [74,75]. It has to be said that not every myopathy is connected with a higher risk of MH, even though each myopathy presents with its own anesthetic and critical care problems [76,77,78]. 

The proportion of patients referred to MH units without a personal/family history of an adverse anesthetic event has increased, with 39.2% as MHS [23]. With the progress in the genetic diagnostic field, more frequently the MH genetic diagnosis, or more often variants of uncertain significance in MH candidate genes, can be only coincidentally found within the investigation of other problems (myopathy, hypotonia, psychomotor development impairment, failure to thrive, etc.). Appropriate interpretation and implementation of this knowledge for non-MH professionals can be difficult, always contact the MH centers. The diagnostic process of MH (see further) takes some time, and till ruling MH out every patient at risk must be treated as susceptible!


**Do not forget: Any patient who has a risk factor or has been given MH-triggering substances might be at risk for MH!**


## 3. Management and Treatment of MH Crisis

### 3.1. Initial ABCDE Patient Approach

The initial assessment of the critically ill pediatric patient should be standardized according to local or international protocols. One of the possible approaches is the ABCDE initial approach recommended by the European Resuscitation Council (ERC) and the European Paediatric Advanced Life Support taskforce (EPALS) [79]. 

### 3.2. Management of MH Crisis—Up-to-Date Guidelines

There were published several guidelines devoted to the management of MH crisis, most recently “Recognizing and managing a malignant hyperthermia crisis: guidelines from the European Malignant Hyperthermia Group (EMHG)” [39] and “Malignant hyperthermia 2020: Guideline from the Association of Anaesthetists” [17]. Relevant information can also be found on the website of the Malignant Hyperthermia Association of the United States (MHAUS) [40]. In recommendations can be found checklists examples, their usage was proven to improve overall performance [80]. In addition, many workplaces commonly have their adapted internal guideline for managing MH crisis, incorporating locally specific information (name of recommended infusions, drugs, availability of dantrolene, phone numbers, etc.). It is very important to be aware of its existence at own workplace and also regularly train critical situation management. The rarity of MH makes it ideal for simulation training, for example, with virtual patients [81], or in high-fidelity simulations [82,83].

The crucial point in the management of the MH crisis is to start treatment as soon as possible. The key to making the diagnosis of MH in a timely manner is to be aware of its possibility whenever triggering agents are used and to have an appropriately tuned index of suspicion [17]. Anesthesiologists are trained for the possibility of MH crisis development whenever the MH triggers were used, whereas, for intensivists or other pediatric specialists, MH can appear so rare and so strongly connected with anesthesia only, that they might misdiagnose it. The later the proper treatment is started, the more severe *complications* can develop and increase the risk of mortality.

#### 3.2.1. Elimination of MH Triggers

Elimination of the triggering agents is crucial and must be the first step in the treatment of the MH crisis [17,39,40]. Turn off and remove the vaporizer or AnaConDa^TM^ system. Set FiO_2_ to 100% oxygen at maximum flow. Also newly if available, the activated charcoal filters (ACF) should be placed on the expiratory and inspiratory limb of the circuit to adsorb volatile anesthetics [17,18,84]. Increase the minute ventilation (2–3 times higher than normal) as it is essential to breathe out the excessive amount of CO_2_ originated from muscular hypermetabolism. Declare an emergency and call for help, for further management of MH crisis is dantrolene as well as more personnel needed. Moreover, inform other members of the team about the situation. One should not lose precious time and personnel on changing the anesthetic circuit, carbon dioxide absorber or even the whole anesthesia workstation. Change to non-trigger anesthesia (TIVA) or sedation in PICU. In case of ongoing procedure or surgery, try to quit it as soon as possible. Meanwhile, establish good IV lines with wide-bore cannulas and consider inserting an arterial and central venous line, and a urinary catheter.

#### 3.2.2. Dantrolene

Dantrolene sodium is an effective antidote to MH [85,86]. It inhibits the excessive release of calcium into the muscle cell. The mortality of MH reached up to 60–70% before the introduction of dantrolene [87,88], and sadly stays over 50% even in modern times in countries, where dantrolene is not readily available [24]. The sooner the patient gets dantrolene, the higher chances for a better outcome. Treatment of MH episodes with intravenous dantrolene has resulted in a reduction in the expected mortality due to MH to as low as 1.4–5%.

The pharmacokinetics of dantrolene is similar in both children and adults The dose of the intravenous initial bolus of dantrolene is usually 2.5 mg/kg, in some guidelines the initial bolus dose is recommended rather within a range for pragmatic reasons when preparing, 2–2.5 mg/kg, or 2–3 mg/kg, respectively [15,17]. However, the traditional formulation of dantrolene is the opposite of an “easy to prepare-quick to administer” drug. The 20 mg vial must be reconstituted with 60 mL sterile water, use a 50 mL syringe filled with up to 60 mL. Because of poor water solubility, it takes quite a long time to dissolve, therefore the more hands the better. Administer each syringe as it is prepared, rather than waiting for the whole dose of the initial bolus to be ready. An example of the possible initial dantrolene bolus preparation in children is shown in Appendix A. The initial bolus administration of dantrolene means quite a big volume of fluids, especially for the smaller children under 10 kg of body weight. This has to be taken into account within the overall fluid management in PICU. An alternative to the traditional formulation is Ryanodex^®^, another form of dantrolene. There is 250 mg of dantrolene in the Ryanodex^®^ vial and it can be rapidly dissolved in 5 mL of sterile water. Unfortunately, it is not yet registered in Europe, and it is quite expensive.

Boluses of dantrolene should be repeated every 10 min until meeting the treatment goal (EtCO_2_ < 6 kPa with normal minute ventilation and body temperature < 38.5 °C) [15,17,40]. Stop giving dantrolene once the treatment goal is reached, but keep in mind the risk of further worsening of even recrudescence [42,43]. There is no maximum dose, the dose of 10 mg/kg per day can be exceeded if needed, but if dantrolene is not helping, consider alternative diagnoses [15,17,39,40]. The continuous infusion of dantrolene or prophylactic dantrolene administration is not recommended any more due to the high incidence of thrombophlebitis (dantrolene is an alkaline solution with pH 9.5 and with poor water solubility) and dose-dependent muscle weakness [89,90]. 

Every anesthesiologist and intensivist must know where to find dantrolene at one´s workplace or where to call for getting it as soon as possible. Because of patient safety and according to the nowadays recommendations, dantrolene should be available at every workplace where MH triggers are given to the patients [15,57,91]. MHAUS recommends that centers stock a minimum of 36 vials of generic dantrolene or three vials of Ryanodex^®^ [92]. According to the EMHG, 36 vials of dantrolene should be immediately available with a further 24 vials available within one hour [15]. However, the real dantrolene availability might differ among many countries.

#### 3.2.3. Cooling

Because the MH crisis is a hypermetabolic whole body state, the body temperature can rise to an extreme value (often even over 42 °C). Therefore, after the recognition of MH, elimination of triggers and administration of dantrolene is the active patient cooling undoubtedly crucial part of the management of MH crisis. Any kind of possible cooling method can be used, starting from the most quickly accessible ones. 

There is a wide range of body temperature control devices, which are classified into two types: noninvasive, which act on a body surface (blankets or pads through which cold water or air circulate) or invasive (intravascular catheters, infusion of ice-cold IV fluid, peritoneal lavage, esophageal or transnasal devices, or extracorporeal circulation) [93]. Endovascular cooling (e.g., Thermoguard) and gel-adhesive pads (e.g., Artic Sun) provide more rapid hypothermia induction and more effective temperature maintenance compared to water-circulating cooling blankets Servo-controlled devices are more effective in controlling body temperature [94]. Children have a larger body surface in proportion to their body weight, as compared to adults. This fact could explain that skin cooling devices are more effective in children. In addition, invasive devices cannot be used in children at a very young age [93]. 

Continuous monitoring of body temperature is an absolute must-have—not only to be aware of ongoing hyperthermia but also to avoid the opposite extreme, hypothermia, caused by excessive cooling, especially, in babies and small children. Monitoring of core temperature is much more reliable than peripheral skin temperature, as it is not altered by skin perfusion. Core temperature can be measured at the tympanic membrane, distal esophagus, and nasopharynx, a bit less accurately also at the rectum and bladder.

#### 3.2.4. Treatment of Other Consequences Manifesting during the MH Crisis 

With the elimination of triggering agents and administering the dantrolene, we are trying to stop the lethal vicious circle, but we also need to treat the MH consequences. Obtain samples for measurement of K^+^, CK, arterial blood gases, myoglobin, and glucose. Check renal and hepatic function and coagulation. Because the treatment of other consequences is the same as for many other reasons besides MH leading to them, it will be described in less detail as in the previous sections, referring to the overall knowledge and practice of pediatric intensivists. The special considerations due to the MH crisis are of course mentioned.

##### Acidosis

The high production of CO_2_, if not exhaled, results in respiratory acidosis, followed by lactate buildup on behalf of anaerobic cell metabolism and therefore metabolic acidosis arises. Hence a combined acidosis usually occurs. Treatment of acidosis is primary via hyperventilation, but the threshold for the administration of sodium bicarbonate is low indeed. If the arterial pH is < 7.2 despite hyperventilation, give intravenous sodium bicarbonate in dose 1–2 mmol/kg [17,40]. 

##### Hyperkalemia

Hyperkalemia results from massive muscle cell damage during the MH crisis. The hyperkalemia treatment should be initiated with glucose and insulin infusion, do not use insulin in IV boluses in children [17]. The example of treatment for pediatric patients according to MHAUS: 0.1 units regular insulin/kg IV and 2 mL/kg of 25% dextrose (% in formulation not important) [40]. Intravenous calcium to protect the myocardium is normally the first-line drug when treating hyperkalemia, but in the case of an MH crisis, it is not recommended in general, as there is some evidence that the influx of extracellular calcium contributes to the calcium overload of the myoplasm [17,95]. Calcium might be used in extreme life-treating hyperkalemia [17,39,40]. Normally recommended inhalational salbutamol in the hyperkalemia treatment algorithm also might be problematic due to the ongoing tachycardia within the MH crisis. Sometimes dialysis may be required.

Cave: there is a high risk of cardiac arrest if hyperkalemia stays untreated!

##### Tachyarrhythmia

The most common arrhythmia in MH is tachyarrhythmia. The most prevalent is usually the narrow complex tachycardia with sinus rhythm due to a hypermetabolic state, but ventricular extrasystoles or ventricular tachycardia might occur as well. Amiodarone in a dose of 5 mg/kg can be given. Moreover, short-acting beta blockers, e.g., esmolol or propranolol, in form of intravenous infusion could be used to stabilize the rhythm if needed. Treatment with calcium blocker drugs should be avoided. Severe cardiovascular instability with hypotension might occur if the treatment of the MH crisis is not extensive enough. An external defibrillator should be available in case of need.

##### Myoglobinuria and Acute Kidney Injury 

Higher levels of serum and urine myoglobin should be anticipated when treating the MH crisis. Aim for the urine output >2 mL/kg/hod to prevent acute kidney injury, this may require an increase in fluid management and diuretics. Urine alkalization is controversial, but it might be a positive side effect as myoglobin precipitates less out of alkaline urine leading to a lower probability of AKI development [17]. Dialysis may be required.

##### Disseminated Intravascular Coagulopathy

The occurrence of DIC during an MH crisis is associated with poor outcomes [17]. Empirical treatment with plasma and platelet transfusions and coagulation factors cryoprecipitates should be commenced [96,97].

##### Compartment Syndrome

Any patient who develops myoglobinuria should be monitored for the development of compartment syndrome [17]. The diagnosis is clinical, its detection might be more complicated in sedated and smaller children. Therefore, an active search for deterioration of peripheral pulses, SpO_2_, swelling and pain should be part of the monitoring. In case of high suspicion, the compartmental pressures should be measured. The treatment is fasciotomies.

##### Continuous Monitoring, Laboratory, and Other Management

According to guidelines, monitoring the patient with suspected MH crisis in a critical care environment should be for a minimum of 24 h [17,39]. The continuous monitoring of at least electrocardiography, non-invasive blood pressure, SpO_2_, EtCO_2_, diuresis and temperature should be the routine in PICU, a modern wireless monitoring system can be used [98]. Consider inserting an arterial and central venous line, and a urinary catheter if not already inserted. Beneficial is a urinary catheter with thermometer—for monitoring both the temperature and the diuresis. The more severe MH episodes require invasive monitoring [99]. Assessment, prevention, and management of key aspects for the comprehensive critical care of infants and children should be treated according to local protocols and international guidelines [100]. Reassess the patient repeatedly, as there is a risk of recrudescence [42,43]. Laboratory blood testing focus on levels of K^+^, CK, arterial blood gases, myoglobin, glucose, renal and hepatic functions, and coagulation. Do not forget that CK may not peak up for up to 24 h after an MH event [101]. Every suspected MH crisis should also be referred to the MH diagnostic center and the patient, and his family should be informed about the MH risk and the need to mention this in case of admission to the hospital.

## 4. Follow-Up MH Diagnostics 

### 4.1. Referring Patient to the MH Center

MH is a rare disease and therefore not every hospital can have enough expertise to manage the diagnostics. In developed countries exist MH centers or MH units, which are providing and managing diagnostics of MH. Therefore, referring every patient with a suspected MH episode or a person newly indicated to be at risk for MH, to an MH center is very important and should be the first step in follow-up management of MH. The MH centers are associated under the MHAUS in the United States or under the EMHG in Europe and some other world countries, contact details can be found on their websites [102,103]. Some of these centers also provide 24/7 Hotline service in case of needed help with managing the outgoing MH crisis.

### 4.2. Testing for MH

Based on the results of testing, the patient is called MH susceptible (MHS) or MH negative (MHN). If the patient is MHS, all blood relatives are at risk of MH and should be tested as well. All patients who are diagnosed to be at increased risk of MH are advised to obtain a Medic Alert or similar tag.

There are two types of testing for MH: genetic testing and muscle biopsy.

The MH diagnostic algorithm has changed fundamentally in the past several years, mainly due to the progress in genetics. Whereas before 2015, the MH diagnosis could not be done without a positive muscle biopsy (at least in one member of the family) [104], after the issuance of a new guideline [105] in 2015 the genetic diagnosis of MH was significantly moved to the forefront. But even though, genetic testing for MH susceptibility has now become widespread, it still does not replace the in vitro contracture tests (the muscle biopsies) [16]. 

#### 4.2.1. Genetic Testing

Genetic testing is provided from a peripheral venous blood sample. There are several molecular genetic methods used, depending on the possibilities of genetic laboratories all around the world. The Sanger Sequencing has been nowadays replaced with a more modern and efficient technique of Next Generation Sequencing (NGS) of genes of interest, whole exome, or even whole genome.

As mentioned in the introduction part, MH is autosomal dominant. Genes causally connected with MH are *RYR1*, *CACNA1S* and more recently also *STAC3* [2,3,4,5]. The detection of a pathogenic diagnostic variant in one of these genes is nowadays enough for calling a patient MHS, even without the muscle biopsy [105]. The positive screening in the family is then done by scoring. However, not every detected variant is pathogenic, there are five variant categories—“pathogenic,” “likely pathogenic,” “uncertain significance,” “likely benign,” and “benign” [106]. The criteria for calling a detected variant “pathogenic” in sense of MH are rather strict [107,108]. Despite this, the list of variants exploitable for genetic testing is growing every year [109]. Unfortunately, in about half of MHS patients, the genetic background and its association with MH are not certain yet. Therefore, genetic testing might not lead to MH diagnosis in each patient or family and MH cannot be ruled out based on genetic testing alone and the patient must undergo a muscle biopsy.

#### 4.2.2. Muscle Biopsy

Muscle biopsy for MH diagnostics means an invasive surgical procedure, when the muscle sample is taken and then straight after tested with Caffeine Halothane Contracture Test (CHCT), or the European version In vitro Contracture Test (IVCT). CHCT/IVCT has been considered the gold standard of MH diagnostics for decades as it has very high sensitivity, and specificity [110]. The biopsy can be performed in trigger-free general anesthesia or neuraxial or peripheral nerve blockade. It is not a simple needle muscle biopsy; the muscle sample is within a few (3–5) centimeters. The muscle sample is taken from the musculus quadriceps femoris, either vastus medialis or lateralis. Due to the needed size of the muscle sample, the biopsy is not performed in children under 30 kg of body weight or less than 10 years of age, in this case, parents can be tested instead. The vital muscle sample is exposed to the triggers of MH, both halothane and caffeine in set concentrations. According to the test protocol, the muscular reaction is then measured and evaluated due to the protocol criteria. Based on the results, the patient is called MHS or MHN. Unfortunately, CHCT/IVCT is very invasive in comparison to genetic screening from a blood sample, but nowadays remains the only way how to exclude MH.

## 5. Conclusions

This comprehensive narrative review on MH in PICU brought together relevant and recent information aiming to filling the knowledge gap about MH in non-anesthesiologists. MH is a rare hereditary disease triggered by substances used not only during anesthesia but also in PICUs. MH can also be rarely triggered by non-pharmacological triggers. The MH genetic diagnosis can be only an incidental finding within the investigation of other problems. MH crisis is a life-threatening severe hypermetabolic whole-body reaction. The most typical symptoms of MH are hypercapnia, tachycardia, hyperthermia, and muscle rigidity. MH must be also considered as a possible cause of rhabdomyolysis or hyperpyrexia in PICU. Recrudescence or delayed onset of MH crisis was previously described. The cornerstone of successful treatment is to start as soon as possible, the sooner, the better patient outcomes. Eliminate the triggers, administer dantrolene, commence cooling and treat adverse consequences. Every suspected MH crisis should also be referred to the MH diagnostic center, and the patient and his family must be informed. 

## Figures and Tables

**Table 1 children-09-01692-t001:** Differential diagnosis of malignant hyperthermia.

Insufficient anesthesia, analgesia, or both
Insufficient ventilation or fresh gas flow
Equipment malfunction or failure (tracheal tube, ventilator, etc.)
Elevated end-tidal CO_2_ due to laparoscopic surgery
Anaphylactic reaction
Infection or septicemia
Pheochromocytoma
Thyroid crisis
Cerebral ischemia
Neuromuscular disorders
Exertional heat stroke, exertional rhabdomyolysis
Ecstasy or other dangerous recreational drugs
Neuroleptic malignant syndrome
Serotonin syndrome

Adapted from published guidelines [17,39,40].

**Table 2 children-09-01692-t002:** Malignant hyperthermia vs. Neuroleptic malignant syndrome vs. Serotonin syndrome.

	MH	NMS	SS
Cause	genetic predisposition, pharmacological triggers: inhalational anesthetics, succinylcholine	antidopaminergic medication: antipsychotics—haloperidol, fluphenazine, clozapine, risperidone; antiemetics—metoclopramide, promethazine	serotonergic medication: SSRIs (sertraline, escitalopram); MAOIs (selegiline, phenelzine); SNRIs; tricyclic antidepressants; other (ondansetron, triptans, linezolid)
Common clinical presentation	altered mental state, increased muscle tone, and autonomic instability (hyperthermia, diaphoresis, tachycardia, hypertension)		
Difference in clinical presentation	hypercapnia	hyporeflexia, extreme “lead-pipe” muscle rigidity followed by massive myoglobinuria	hyperreflexia, neuromuscular hyperactivity (clonus, ataxia, tremors)
**Time to develop**	minutes to hours	days	hours
**Treatment**	dantrolene	bromocriptine, dantrolene	cyproheptadine

Adapted from Picmonic [72]. MH = malignant hyperthermia; NMS = neuroleptic malignant syndrome; SS = serotonin syndrome; SSRIs = selective serotonin reuptake inhibitors; MAOIs = monoamine oxidase inhibitors; SNRIs = serotonin and norepinephrine reuptake inhibitors.

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
