# Peer review of "Malignant Hyperthermia in PICU—From Diagnosis to Treatment in the Light of Up-to-Date Knowledge"

_children, 2022, doi:10.3390/children9111692_

Round 1
Reviewer 1 Report
Review Report Form
Manuscript ID: children- 1922123
· Summary
o The background of this manuscript describes malignant hyperthermia (MH) and the associated genes, prevalence in a young population, and the importance of intensivists detecting and treating MH. The aim of the article is to summarize current knowledge on MH, including the detection, treatment, and management of MH in the PICU. This manuscript contributes to the literature by providing an organized summary of recognition and management of MH and specifically addresses treatment in a pediatric population presenting in the intensive care unit. My overall impression is that this review should be considered for publication after several questions regarding the significance and novelty of the work are addressed by the authors.
· Review
o The aim of the review is clear but needs more support in the introduction and conclusion. The introduction does not help the reader understand how a summary on MH in the intensive care setting adds to the current literature. A valuable part of the review includes the sections on differentials to consider in the PICU, so mention of reviewing differentials in the introduction may engage an intensivist earlier during the review. A few questions that can be addressed, which may better support the aim, include:
§ Where are there gaps in our knowledge of the management of MH in the PICU?
§ What are the recent concerns of management of MH in the PICU?
§ What are the identified differences between management of MH by anesthesiologist compared to intensivist in the critical care setting?
§ What are some examples of changes in the guidelines for recognition and treatment that have been made over recent years?
§ In the conclusion section, how can the significance of this review be addressed clearly?
o Overall, the paper is easy to understand and organized; however, there are several grammatical errors that would benefit from careful review by an editor. Additionally, there is inconsistency in the use of malignant hyperthermia versus MH throughout the paper (Lines 45, 117, 126, 209, 245, 489). When re-introducing the term in a paragraph, the abbreviation should be subsequently re-introduced.
· Specific Comments:
o Line 117/Table 1- I appreciate a table for the differential diagnosis; however, I recommend removing the bullet points and shifting the text left in the table to make the table visibly appealing. The title of the table should not use an abbreviation.
o Line 60- “MH is more often detected in children and young adults” Which population is being compared to children and young adults?
o Line 75, A title for subject 2.1 is needed.
o Line 108-110, MODS and MOF only used here, so may not need abbreviations
Author Response
- The background of this manuscript describes malignant hyperthermia (MH) and the associated genes, prevalence in a young population, and the importance of intensivists detecting and treating MH. The aim of the article is to summarize current knowledge on MH, including the detection, treatment, and management of MH in the PICU. This manuscript contributes to the literature by providing an organized summary of recognition and management of MH and specifically addresses treatment in a pediatric population presenting in the intensive care unit. My overall impression is that this review should be considered for publication after several questions regarding the significance and novelty of the work are addressed by the authors.
- Review
- The aim of the review is clear but needs more support in the introduction and conclusion. The introduction does not help the reader understand how a summary on MH in the intensive care setting adds to the current literature. A valuable part of the review includes the sections on differentials to consider in the PICU, so mention of reviewing differentials in the introduction may engage an intensivist earlier during the review. A few questions that can be addressed, which may better support the aim, include:
Answer: Dear Reviewer 1, thank You very much for Your comments. We are delighted, that you liked our narrative review. With help of Your comments, we tried to highlight the aim and contribution of this review much more in our manuscript. We rewrote the introduction, hopefully now is more engaging for non-anesthesiologists much sooner.
- Where are there gaps in our knowledge of the management of MH in the PICU?
Answer: The gap is on the side of MH's genetic background and recognition of a person at risk for MH. With the current genetic knowledge and with the developing molecular genetics diagnostic methods, more and more patients are diagnosed as MH susceptible (MHS) without any adverse anesthetic complication, and more frequently the MH genetic diagnosis, or more often variants of uncertain significance in MH candidate genes, can be only coincidentally found within the investigation of other problems (hypotonus, psychomotor development impairment, failure to thrive, etc.). Appropriate interpretation and implementation of this knowledge for non-MH professionals can be difficult. In the progressive genetic era, MH is now less anesthesiology-inclusive and pediatric specialists can meet MH susceptible or risk patients much more often. The better information we will provide to the professionals outside of anesthesia specialty, the better early detection of patients at risk, also the faster recognition of MH crisis and commencing the right management, and subsequently fewer complications and better patient outcome. Also, to the best of our knowledge, this is the first narrative review article devoted to MH from the pediatric intensivist´s point of view. We also added the explanation of our aim and the knowledge gap to the article (at the end of the introduction part).
- What are the recent concerns of management of MH in the PICU?
Answer: The answer to this point relates to the answer above, the concerns are that we simply do not know yet the genetic background of all MHS patients. The key to diagnosing an MH crisis is to be aware that it can develop in anybody anytime the triggering anesthetics are administered, or as already mentioned very rarely even without them. But one must be tuned to the possibility of MH and take MH into account in the differential diagnosis. It is not only dantrolene, which saves lives – early warning, diagnosis, and prompt effective therapies are crucial for MH patients to survive, as proven in China, where the availability of dantrolene is very low and yet they were able to reduce the MH mortality (Gong, 2021).
- What are the identified differences between management of MH by anesthesiologist compared to intensivist in the critical care setting?
Answer: The main difference is that anesthesiologist is trained for the possibility of MH, whereas for intensivists or other pediatric specialists MH can be so rare and so strongly connected with anesthesia only, that they might miss it. We added this point to the article (section 2. Topics of concern regarding to MH in PICU).
- What are some examples of changes in the guidelines for recognition and treatment that have been made over recent years?
Answer: The recommendation for the use of activated charcoal filters (ACF) if available (lines 374-376). Dantrolene boluses are recommended now, continuous infusion of dantrolene or prophylactic dantrolene administration is not recommended any more due to the high incidence of thrombophlebitis (lines 410-418). An alternative to the traditional formulation is Ryanodex® (lines 352-355). And the biggest difference is in the diagnostic process of MH and the much more important role of genetic diagnostics nowadays (described more in sections 4.2. Testing for MH and 4.2.1. Genetic testing).
- In the conclusion section, how can the significance of this review be addressed clearly?
Answer: Thank You very much for this comment, we rewrote the conclusion part to make it more clearly.
- Overall, the paper is easy to understand and organized; however, there are several grammatical errors that would benefit from careful review by an editor. Additionally, there is inconsistency in the use of malignant hyperthermia versus MH throughout the paper (Lines 45, 117, 126, 209, 245, 489). When re-introducing the term in a paragraph, the abbreviation should be subsequently re-introduced.
Answer: Thanks for Your detailed review. We tried to focus more on the consistency of used terminology, and we corrected the terminology throughout the manuscript.
- Specific Comments:
- Line 117/Table 1- I appreciate a table for the differential diagnosis; however, I recommend removing the bullet points and shifting the text left in the table to make the table visibly appealing. The title of the table should not use an abbreviation.
Answer: Thanks for this comment, it is now corrected.
o Line 60- “MH is more often detected in children and young adults” Which population is being compared to children and young adults?
Answer: Ibarra et al in their recent publication focused on An Assessment of Penetrance and Clinical Expression of Malignant Hyperthermia multicenter case–control study used data from 125 MH pedigrees between 1994 and 2017 which were collected from four European registries and one Canadian registry. Age distribution in probands was positively skewed (fig. 3), with a median age of 12 yr and an overwhelming majority being younger than 33 yr old. It was also described in older studies as well. We rephrased the end of introduction according to your comments, so hopefully now we answered your comment in a better way.
o Line 75, A title for subject 2.1 is needed.
Answer: Thanks for this comment, it is now corrected.
Line 108-110, MODS and MOF only used here, so may not need abbreviations
Answer: Thanks for this comment, it is now corrected.

Reviewer 2 Report
Reviewer comments
It is well writing Review on Malignant Hyperthermia in PICU – from diagnosis to treatment in the light of up-to-date knowledge." the review dealt with an important topic and with useful scientific content with accepts for publication.
Comments:
1- It is well writing study on this topic.
2- The design and performance of this study were sufficiently documented.
3- The reference list covers the relevant literature adequately.
4- It is a highly interest to a general audience.
Author Response
Thank You very much for Your very kind review. We appreciate it deeply.

Reviewer 3 Report
The manuscript is well written and should be accepted for publication
Author Response

(The authors gave the same response as above.)

Round 2
Reviewer 1 Report
I appreciate the responses to my comments by the authors. The authors did a great job addressing my questions and comments on their manuscript. I have no further suggestions/comments for the authors. I recommend this manuscript is published in its present form.